# Item-level psychometrics of the Ascertain Dementia Eight-Item Informant Questionnaire

Yeajin Ham[1], Suyeong Bae[1], Heerim Lee[1], Yaena Ha[1], Heesu Choi[1], Ji-Hyuk Park[2], Hae Yean Park[2], Ickpyo Hong[2]*

1 Department of Occupational Therapy, Graduate School, Yonsei University, Wonju-si, Gangwon-do, Republic of Korea, 2 Department of Occupational Therapy, College of Software and Digital Healthcare Convergence, Yonsei University, Wonju, Gangwon-do, South Korea

* ihong@yonsei.ac.kr

## Abstract

The aim of this study is to evaluate the item-level psychometrics of the Ascertain Dementia Eight-Item Informant Questionnaire (AD-8) by examining its dimensionality, rating scale integrity, item fit statistics, item difficulty hierarchy, item-person match, and precision. We used confirmatory factor analysis and the Rasch rating scale model for analyzing the data extracted from the proxy versions of the 2019 and 2020 National Health and Aging Trends Study, USA. A total of 403 participants were included in the analysis. The confirmatory factor analysis with a 1-factor model using the robust weighted least squares (WLSMV) estimator indicated a unidimensional measurement structure ($\chi^2$ = 41.015, $df$ = 20, $p$ = 0.004; root mean square error of approximation = 0.051; comparative fit index = 0.995; Tucker–Lewis Index = 0.993;). The findings indicated that the AD-8 has no misfitting items and no differential item functioning across sex and gender. The items were evenly distributed in the item difficulty rating (range: −2.30 to 0.98 logits). While there were floor effects, the AD-8 revealed good reliability (Rasch person reliability = 0.67, Cronbach's alpha = 0.89). The Rasch analysis reveals that the AD-8 has excellent psychometric properties that can be used as a screening assessment tool in clinical settings allowing clinicians to measure dementia both quickly and efficiently. To summarize, the AD-8 could be a useful primary screening tool to be used with additional diagnostic testing, if the patient is accompanied by a reliable informant.

## Introduction

According to the World Health Organization (WHO), the world population belonging to the age group 65 years and above is expected to reach 1 billion by 2020 and 2 billion by 2050 [1]. As the birth rate decreases and life expectancy increases, demographic changes emerge worldwide, resulting in a society that is growing old and has varied socioeconomic impacts [2]. Due to this aging, geriatric chronic diseases, such as Alzheimer's and other neurodegenerative diseases, and cardiovascular diseases, are rising exponentially [3]. Furthermore, the number of patients with dementia is expected to increase to 78 million by 2030 and 139 million by 2050,

**Data Availability Statement:** All files are available from the National Health and Aging Trends Study database. https://nhats.org/researcher/data-access.

**Funding:** This work was supported by the Ministry of Education of the Republic of Korea and the

National Research Foundation of Korea (NRF-2021S1A3A2A02096338) and the Institute of Convergence Science (ICONS) at Yonsei University. The funders had no role in study design, data collection and analysis, decision to publish, or preparation of the manuscript.

**Competing interests:** The authors have declared that no competing interests exist.

corresponding to the increase in the elderly population [1]. According to the Center for Disease Control and Prevention (CDC) data, dementia places a great socioeconomic burden on the patient and their family resulting from decreased memory, attention, reasoning, judgment, and problem-solving ability [4].

Given that dementia is not an inevitable sequela of aging, early detection and appropriate treatment to delay its onset are crucial [5]. The most prominent symptoms of dementia are memory and executive dysfunction, which are accompanied by other neuropsychiatric symptoms (NPS) and decreased ability and speed in the execution of activities of daily living (ADLs) [6]. A study reported that over 90% of patients with dementia had at least one NPS, leading to severe deterioration of the patient's quality of life and was strongly associated with caregiver burden [7]. Hence, early detection may help reduce the family suffering and social costs through appropriate treatment and management [8].

A systematic review described the following assessment tools used to screen for dementia in a community setting: Montreal Cognitive Assessment (MoCA), Addenbrooke's.

Cognitive Examination-III (ACE-III), Saint Louis University Mental Status, and Rapid Cognitive Screen [9]. However, these assessment tools require >10 minutes for administration, which limits their use as a screening tool in clinical settings wherein the clinicians are required to complete the initial assessments quickly.

The Ascertain Dementia Eight-Item Informant Questionnaire (AD-8) is a dementia screening tool demonstrating excellent correlation and concurrent validity to other screening tools and can be used as an adjunct to other tools [10, 11]. The AD-8 consists of eight items to help clinicians or healthcare providers quickly detect cognitive impairment [12]. It has a short administration time, an average of 3 minutes; the constituent items have a dichotomous response category (yes or no) and can be divided into four conceptual domains–memory, endurance, execution ability, and complex functions [10].

It is assumed that the AD-8 is a valuable tool for screening possible cognitive impairment and can be applied to clinical settings. However, so far, no study has evaluated its construct validity including item-level psychometrics. Construct validity refers to how well the evaluation tool measures the variable being evaluated [13], whereas Rasch analysis is applied to study the feasibility of an evaluation tool [14]. Rasch analysis helps establish the internal consistency and reliability of each item constituting the test [15], focusing on each question [16]. To the best of our knowledge, this methodology has not yet been applied to inspect the psychometric properties of the AD-8. Therefore, this study aimed to apply Rasch analysis to confirm the psychometrics of the AD-8. We evaluated dimensionality, rating scale integrity, item fit statistics, item difficulty hierarchy, item-person match, and precision of the AD-8. Additionally, we checked for the presence of any differential item functioning in age and sex, or floor and ceiling effects.

## Materials and methods

### Study settings

The data for this analysis was obtained from the proxy versions of the National Health and Aging Trends Study (NHATS) conducted in 2019 and 2020 (round 9, round10) [17, 18]. The NHATS is a health-related survey carried out annually since 2011 for older adults aged 65 years and above who are Medicare beneficiaries in the United States. The NHATS gathers critical information about the respondents, including sociodemographics, physical and cognitive functions, health status, and medication; a proxy, typically a family member, can complete the survey when the participant is not available due to cognitive impairment or health issues.

We collected the observations responding to the AD-8 and excluded entries with any missing data of the questions from NHATS data of rounds 9 and 10. This study was exempt from ethical clearance by the local Institutional Review Board in Korea as the study utilized publicly available de-identified data from the NHATS.

## AD-8 questionnaire

Cognitive function in the NHATS is measured by the Washington University Dementia Screening Interview, also known as the AD-8, a brief and simple screening tool that distinguishes individuals with potential dementia or mild cognitive impairment [11]. AD-8's eight questions identify the difficulties experienced by the individuals in the last several years due to memory and cognitive impairment to indicate the risk of cognitive decline [19]. The AD-8 is an informant-based assessment and can be completed by patients, caregivers (spouse, child, etc.), or practitioners; it can be administered in person or by phone. The average administration time is <3 minutes and the interviewer is required to undergo minimal training. Additionally, the AD-8 is reported to be appropriate for primary and tertiary health care settings and community settings [20, 21].

The AD-8 consists of a 3-point rating scale, including the following responses: (1) Yes, a change, (2) No, no change, and (3) N/A, do not know. When analyzing the response using Rasch analysis, we collapsed the rating scale into a dichotomous response category: 1-point is given for each "Yes, a change" and 0 for "No, no change" and "Do not know." A total score higher than 2 points indicates possible cognitive impairment, which requires a more precise dementia test [22]. A previous study reported that the AD-8 offers good reliability and validity with a Cronbach's alpha of 0.84 (95%, confidence interval (CI) = 0.80–0.87) [11].

## Data analyses

**Unidimensionality.** In the Rasch model, one of the core assumptions is a unidimensional measurement structure [23]. Before calibrating the AD-8 using the Rasch model, we conducted a confirmatory factor analysis (CFA) with a one-factor model to assess the unidimensionality assumption for the AD-8. The Chi-square test was used to evaluate the overall model fit, wherein non-significance ($p > 0.05$) indicated that the model fits the data [24]. Similarly, a root mean square error of approximation (RMSEA) of <0.08 is required for a good fit, whereas RMSEA <0.06 is considered an excellent fit [25]. Also, a comparative fit index (CFI) of >0.95 and Tucker–Lewis Index (TLI) of >0.95 are considered good fit [25]. In categorial data, the appropriate estimation methods are weighted least squares (WLS) or robust weighted least squares (WLSMV) [26]. We utilized the WLSMV estimator accoriding to the PROMIS group guideline [25]. Mplus version 8.4 (LosAngeles, CA, 2012) was used for the CFA.

**Local independence.** We examined the local independence of all eight items on the AD-8. Local independence influences the unidimensionality of an instrument and should be checked before Rasch analysis [27]. When residual correlations among the test items are greater than 0.2, those items are considered to violate the local independence assumption [25].

## Rasch analysis

**Rating scale analysis.** We assessed the rating scale of the AD-8 with the Rasch rating scale model using the following three crucial criteria: 1) the number of observations is greater than10 for each rating scale category; 2) rating score measures (category measures) advanced from the lowest to the highest rating score; 3) outfit mean square value (MnSq) less than 2.0 per rating scale category is required [28].

**Item fit statistics.** These statistics identify the misfitting items in the Rasch model [16]. We followed Wright's guidance to assess the item fit statistics; the ideal item MnSq is 1.0 with an infit and outfit range of 0.6–1.4. Additionally, the standardized Z-value ($Z_{std}$) should be less than 2.0 (Linacre & Wright, 1993).

**Item difficulty hierarchy.** Item difficulty hierarchy is used to calculate the number of people who succeeded at an item from the total number of people who attempted it [16]. Item difficulty hierarchy in this study was analyzed in ascending order. A high measured value means that the item was difficult.

**Differential item functioning.** Subsequently, we performed differential item functioning (DIF) analysis to estimate the invariance and stability of item hierarchy among the different sex and age groups. DIF analysis is generally applied for comparing two groups when the expected performance of both groups is different [29, 30]. For example, DIF occurs when participants respond differently to individual scale items based on similar variables, such as sex and age [31]. It is known that cognitive ability is a significant factor for age-related changes [32–34]; according to Thacker et al., cognitive function worsens after the age of 80 years [35]. For this study, we hypothesized that the sex and age of respondents will not affect their responses to all items of the AD-8. The following criteria for determining item DIF were used: a DIF contrast of $|DIF| \geq 0.43$ logits, and a $p$-value of $\leq 0.05$ [23].

**Item-person match.** We also checked the item-person match that indicates the item difficulties and a person's ability on the same interval-logit scale. A logit scale value of 0 is forthwith fixed as the average item difficulty measure for the data. The ceiling effect was accounted for when >15% of the respondents were placed at the maximum extreme score, whereas floor effects were considered when >15% of the respondents had minimum extreme scores [36].

**Precision and reliability.** The instrument's precision represents the capability of the target population with the aimed items in the form of error estimates. Rasch reliability indicates the consistency of the repeatability of the instrument when used for measurements [23]. As per the Rasch model, score precision is a function of the information available and is used to estimate a given score–a given ability level. We determined precision using the person-separation reliability ratio, considering a ratio of >2.0 and Rasch reliability of 0.80 acceptable [37]. All Rasch analyses were conducted Winsteps version 4.7.0.

## Results

### Participants

Fig 1 illustrates the flow diagram of cohort selection. There were 4,977 observations in the NHATS round 9 databases. Among them, 273 participants responded to the AD-8 questionnaire, and after excluding 88 observations that lacked responses to any of these eight questions, 185 observations were finally selected. In the NHATS round 10 databases, we selected 297 participants from 4,389 observations by excluding observations not responding to the AD-8 questionnaire and any missing responses and/or values in sex. After merging the two datasets, 79 duplicated observations were used from recent data (round 10), and a total of 403 participants were analyzed in this study.

Mathematically, a sample size of 250 subjects or more is required in the Rasch model to achieve a definitive item calibration stability (over 99% confidence) [38]. The largest group was formed by 164 people aged ≥90 years (40.69%); 108 people were aged between 85–89 years (26.80%), 72 people aged between 80–84 years (17.87%), 47 people aged 75–79 years (11.66%), and 12 people aged 70–74 years (2.98%). Finally, this study cohort included 85 (21.09%) males and 318 (78.91%) females. More detailed demographic information is described in Table 1.

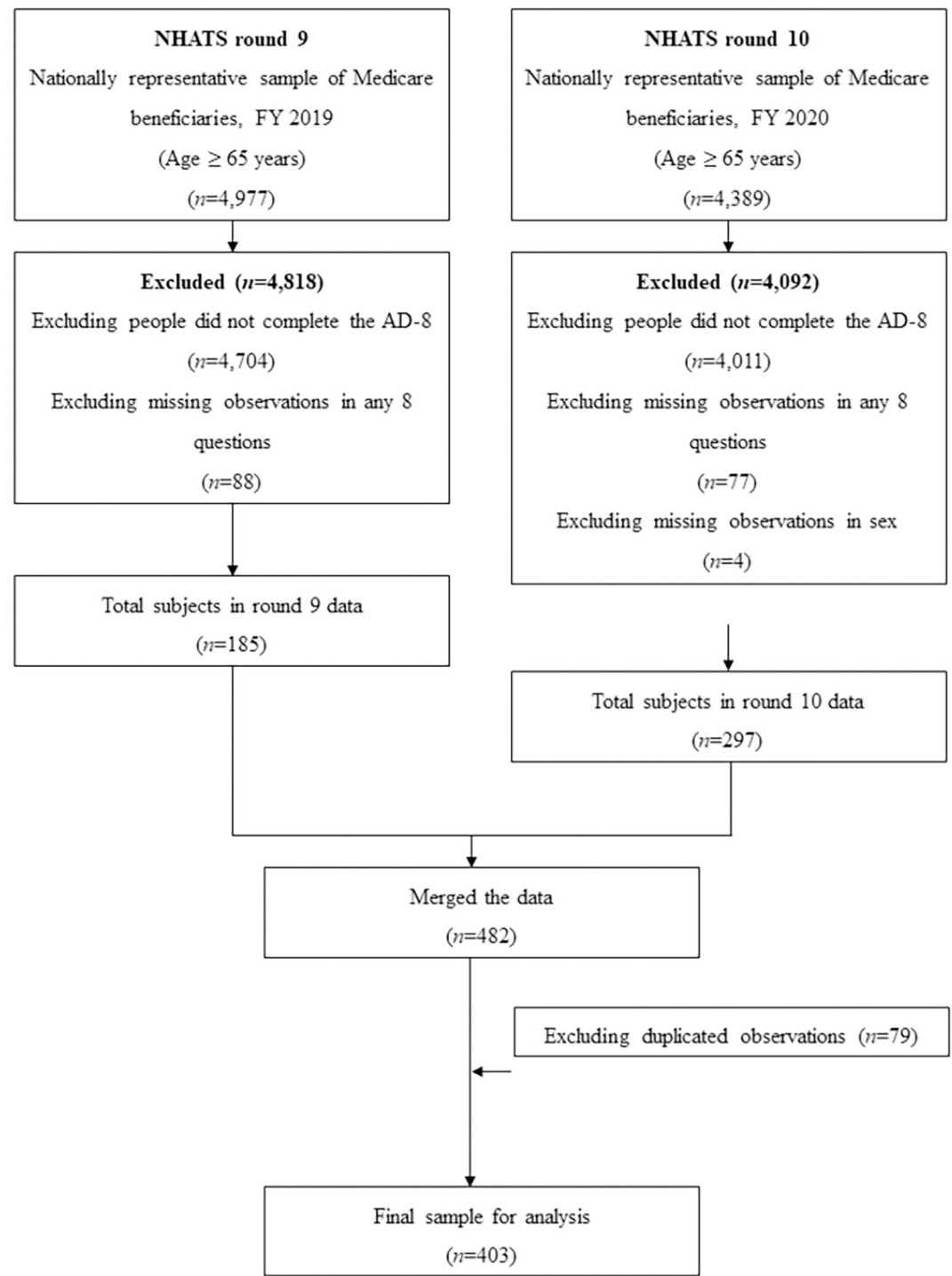

**Fig 1. Cohort selection flow diagram.**

## Unidimensionality

**CFA and local independence.** The CFA with a 1-factor model for the AD-8 is presented in Fig 2. The results of a CFA indicated good model fit based on the configuration validity

**Table 1. The demographic characteristics of the study participants.**

| Variable | $n$(%) |
| --- | --- |
| **Age** | |
| 70–74 | 12 (2.98) |
| 75–79 | 47 (11.66) |
| 80–84 | 72 (17.87) |
| 85–89 | 108 (26.80) |
| Over 90 | 60 (41.96) |
| **Sex** | |
| Male | 85 (21.09) |
| Female | 318 (78.91) |
| **Race** | |
| Non-Hispanic White | 232 (57.57) |
| Non-Hispanic Black | 100 (24.81) |
| Hispanic | 42 (10.42) |
| Others[a] | 29 (7.20) |

[a]Others included American Indians, Asian, Native Hawaiian, Pacific Islander, or mixed.

($\chi^2$ = 41.015, $df$ = 20, $p$ = 0.004; CFI = 0.995; TLI = 0.993; RMSEA = 0.051). These results showed that the factor loadings for all sub-items were higher than 0.5. Subsequently, we examined residual correlations to determine the local independence of the AD-8 and observed that all items scored <0.2. The results of CFA using other estimation methods were shown in S1 Table.

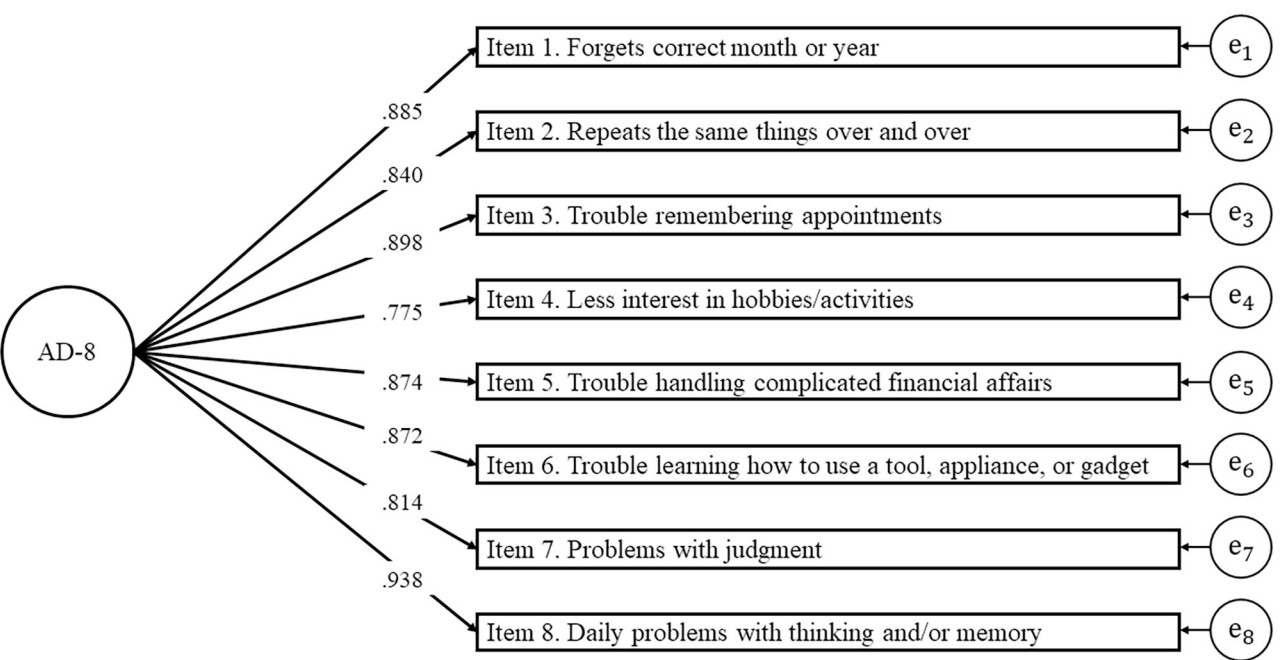

**Fig 2. The diagram of confirmatory factor analysis for the AD-8.**

## Rasch analysis

**Rating scale analysis.** The two-point AD-8 rating scale fulfilled all three essential criteria necessary to employ the Rasch model.

**Item fit statistics and Item difficulty hierarchy.** Table 2 presents the MnSq, $Z_{std}$, and item difficulty hierarchy value of the AD-8. We found that all AD-8 items satisfied the necessary criteria for Rasch item fit. The AD-8 item with the lowest measure was "Problems with judgment," indicated as the most challenging item in the Rasch model, whereas the "Trouble remembering appointments" items had the highest measure, meaning the easiest item in the Rasch model. Most of the items were evenly distributed for difficulty (range: −2.30 to 0.98 logits). The average item difficulty measure of the items "Forgets correct month or year" and "Trouble handling complicated financial affairs" was similar (0.30 logits and −0.09 logits, respectively).

As per the DIF analysis based on age and sex groups (Table 3), no DIF was found according to the sex (female vs. male) and age (70–80 years vs. >81 years) of the respondent.

**Item-person match.** Overall, 48 of 403 (11.9%) patients reported severe cognitive problems and showed a maximum extreme score; 112 (27.8%) patients reported no cognitive problems and had a minimum extreme score. Fig 3 explains the difficulty of items and how well-assessed a person is. Except for item 7, all other AD-8 items were grouped between -1 and 1 logits. In our study, the AD-8 represented a floor effect without a ceiling effect (Fig 3).

**Table 2. Item fit statistics for the Ascertain Dementia Eight-Item Informant Questionnaire (AD-8).**

| Item difficulty | Tool items | Measure | Error | Infit | | Outfit | |
|---|---|---|---|---|---|---|---|
| | | | | MnSq | $Z_{std}$ | MnSq | $Z_{std}$ |
| *Least commonly reported as a problem* | 3. Trouble remembering appointments | 0.97 | .18 | 0.96 | −0.44 | 0.83 | −0.85 |
| ↕ | 8. Daily problems with thinking and/or memory | 0.81 | .17 | 0.78 | −2.54 | 0.59 | −2.67 |
| *Most commonly reported as a problem* | 2. Repeats the same things over and over | 0.66 | .17 | 1.12 | 0.98 | 0.72 | 0.74 |
| | 5. Trouble handling complicated financial affairs | 0.30 | .16 | 0.98 | −0.20 | 0.95 | −0.31 |
| | 1. Forgets correct month or year | −0.09 | .16 | 0.92 | −1.00 | 0.9 | −0.85 |
| | 6. Trouble learning how to use a tool, appliance, or gadget | −0.16 | .16 | 0.98 | −0.24 | 0.91 | −0.76 |
| | 4. Less interest in hobbies/activities | −0.19 | .16 | 2.61 | 2.04 | 0.71 | 0.75 |
| | 7. Problems with judgment | −2.30 | .17 | 1.08 | 0.77 | 0.71 | 0.73 |

MnSq, mean square; $Z_{std}$, standardized z-value

**Table 3. Results of the differential item functioning analyses.**

| Tool item | Sex | | Age | |
|---|---|---|---|---|
| | Females (n = 318) vs. Males (n = 85) | | 70–80 years (n = 59) vs. over 81 years (n = 344) | |
| | DIF contrast | Mantel–Haenszel probability | DIF contrast | Mantel–Haenszel probability |
| **1. Forgets correct month or year** | −0.44 | .424 | 0.53 | .225 |
| **2. Repeats the same things over and over** | −1.05 | .056 | −0.37 | .401 |
| **3. Trouble remembering appointments** | 0.53 | .284 | −0.13 | .771 |
| **4. Less interest in hobbies/activities** | 0.24 | .947 | 0.28 | .515 |
| **5. Trouble handling complicated financial affairs** | 0.04 | .872 | −0.68 | .117 |
| **6. Trouble learning how to use a tool, appliance, or gadget** | −0.04 | .858 | 0.06 | .894 |
| **7. Problems with judgment** | 0.35 | .588 | 0.21 | .627 |
| **8. Daily problems with thinking and/or memory** | −0.33 | .383 | 0.04 | .925 |

DIF, differential item functioning

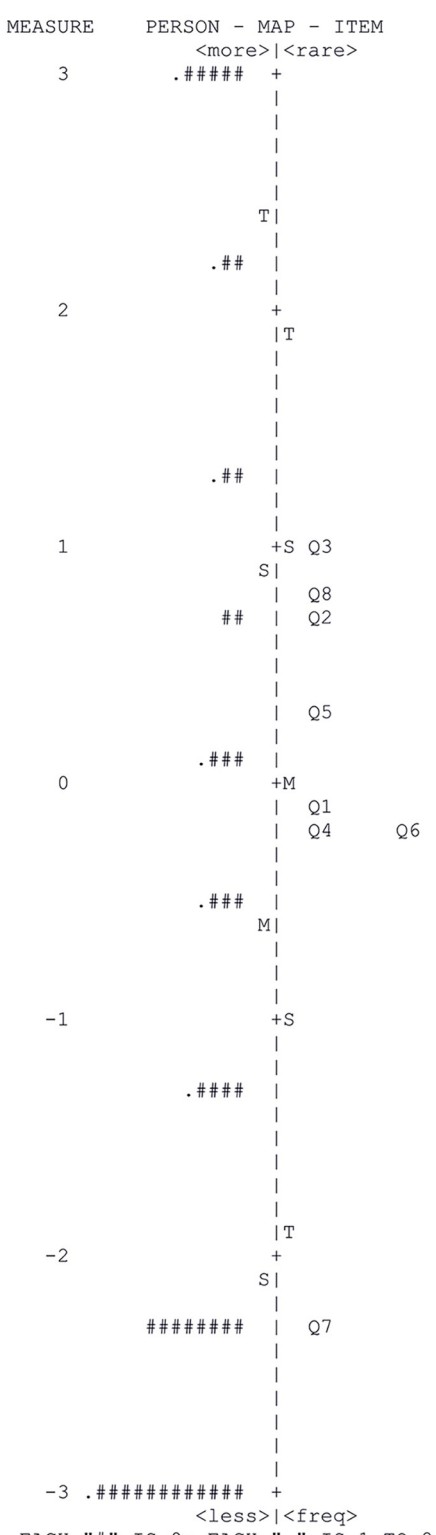

**Fig 3. Rasch person-item map for the AD-8.**

**Precision and reliability and Rasch principal component analysis.** The Rasch person-separation value of the AD-8 was 1.41, the person-strata value was 2.21, the person-reliability value was 0.67, and the Cronbach's alpha for the AD-8 was 0.89. A 62.6% value was observed in the Rasch principal component analysis of the AD-8; this is raw variance explained by the assessment tool. The Eigenvalue of the first contrast was 1.53.

## Discussion

The AD-8 is a widely used and studied clinically-applicable screening instrument to detect dementia. The trend of research on the AD-8 is increasing based on the PubMed search engine (http://www.ncbi.nlm.nih.gov), but the item-level psychometrics of this questionnaire remains indeterminate. This study confirmed the unidimensionality of the AD-8 using CFA, and then examined its construct validity by applying the Rasch model. No misfitting items and no differential items were observed functioning across sex and gender. While we did observe a floor effect from the item-person match, the AD-8 revealed good reliability.

The AD-8 evaluates how dementia symptoms affect ADLs and instrumental ADLs (IADL) rather than memory and cognitive functions. In the prodromal phase of dementia, a decrease in IADL is observed; therefore, IADL impairment is crucial for detecting dementia at an early stage [39]. Moreover, widely used assessment tools, such as MoCA and ACE-III, examine a broad range of cognitive functions, while other performance-based dementia screening tools, such as the Brief Alzheimer Screen and Brief Memory and Executive Test, require appropriate settings for the test [40]. On the other hand, the AD-8 does not require any preparation and could be used for screening dementia quickly and efficiently without any space and time constraints.

The item hierarchy analysis in this study showed that the easier items were related to orientation and memory, while more challenging items were related to executive functions. Verlinden et al. (2016) summarized the hierarchical trajectory of functional decline in dementia–initially, a subjective decline in memory occurs, followed by deterioration in IADL, and lastly, basic ADL independence is lost [41]. Furthermore, another study found that functional recession in the temporoparietal association caused the central executive system dysfunctions [42]. Our results regarding the item hierarchy also found that the memory-related questions were least problematic, while the questions related to IADL and executive functions were mostly reported as a problem.

We also observed floor effects from the Rasch person-item map. The floor and ceiling effects indicate limited content validity [35]; however, they are also associated with higher sensitivity [43]. A previous study of the AD-8 compared four dementia tests, including the AD-8, MMSE, participant subjective memory complaint (SMC), and the informant SMC [44]. The AD-8 had the greatest sensitivity (87.4%) but the lowest specificity (49.4%). This characteristic is supportive of the use of AD-8 as a screening instrument since other screening tests are not diagnostic of dementia but are connected to additional assessment. In line with Morris et al., our results suggest that the floor effects were shown because of the AD-8's high sensitivity and low specificity. Additionally, the floor effect can be explained by considering that the lowest AD-8 score reflected individuals without cognitive function issues; the majority of older adults were not suspected to have dementia.

In the NHATS data, all respondents in AD-8 were proxy. Proxy versions were developed for use in exceptional cases where the patient is mentally or physically not capable of reporting their health-related quality of life [45]. Proxy data may not necessarily accurately reflect the subjective characteristics of patients; however, the use of such data has many advantages, especially for people with mental health problems. Memory loss or dementia is often accompanied

by anosognosia in several people, wherein the use of proxy data may be more accurate than patient-provided data [46]. A recent study compared the correlation of the MoCA and the AD-8 between the patient-reported and proxy versions [47]. The MoCA is a cognitive screening test designed to assist health professionals to detect mild cognitive impairment and Alzheimer's disease (Copyright 2021, Ziad Nasreddine, MD). Denny et al. found a correlation between the proxy version of the AD-8 results and the MoCA scores. Notably, no correlation was found between the patient-reported version of the AD-8 and the MoCA scores [47]. This indicates that when evaluating patients with cognitive impairment, the reliability of the proxy version could be higher than that of the patient-reported version.

In this study, we did not find any misfitted items in the AD-8, i.e., all eight items were appropriate for determining dementia. However, for one of the items, "Less interest in hobbies/activities?" the fit was marginal. Engaging in hobbies and physical or cognitive activities, such as playing golf, craftworks or reading books, has been found to reduce the risk of dementia [48, 49]. However, the proxy respondents may not be able to answer this accurately about the patient's level of interest because it is more subjective than objective. It is already known that proxy respondents are more reluctant to respond to subjective questions, such as patients' feelings or opinions [45]. Therefore, the examiner should pay careful attention when proceeding with this item in the case of the proxy version.

This study had some limitations. First, we used secondary data, which may have caused bias, such as selection or measurements bias and time-lag [50]. Further, we could not get information about the missing value, and it was difficult to analyze data because there was no information on why the respondents did not answer accurately Among the AD-8 respondents in this study, 88 and 77 participants did not complete the assessment in NHATS 2019 and 2020, respectively (missing rates were 32% and 20%, respectively). Additionally, we could not control the answers ("No" and "don't know") that were part of the same scoring system. Nevertheless, secondary data can give broad information that can help to investigate clinical questions. We need transparency and statistical understanding when handling secondary data. Therefore, had we analyzed the data with professional help, we could have clarified better on our clinical research question.

Finally, we checked only sex and age variables when analyzing DIF. Since the data were from the secondary data source, it was not easy to divide it into two different levels for other variables, such as race and educational level. Further research might need to consider other variables besides sex and age. Further statistical analyses are warranted to confirm the criterion validity using a diagnostic gold standard or test-retest reliability tests.

## Conclusion

The Rasch model indicated that the AD-8 has good item-level psychometric properties for older adults aged 65 years and above who are Medicare beneficiaries in the United States. We observed that all eight items fit well and had no DIF in age and sex. The great psychometric properties of the items will allow clinicians to measure dementia in quick and efficient ways. Ultimately, the AD-8 could be a useful primary screening tool to be used with additional diagnostic testing if the patient is accompanied by a reliable informant.

## Supporting information

**S1 Table. Results of the confirmatory factor analysis of two estimation methods.** (DOCX)

## Author Contributions

**Conceptualization:** Yeajin Ham, Suyeong Bae, Yaena Ha, Hae Yean Park.

**Formal analysis:** Suyeong Bae.

**Funding acquisition:** Ji-Hyuk Park.

**Investigation:** Yeajin Ham, Heerim Lee, Heesu Choi.

**Methodology:** Yeajin Ham, Suyeong Bae, Yaena Ha, Heesu Choi, Ickpyo Hong.

**Project administration:** Yeajin Ham, Ji-Hyuk Park, Hae Yean Park, Ickpyo Hong.

**Resources:** Heerim Lee.

**Supervision:** Ji-Hyuk Park, Hae Yean Park, Ickpyo Hong.

**Writing – original draft:** Yeajin Ham, Suyeong Bae, Heerim Lee, Heesu Choi.

**Writing – review & editing:** Yeajin Ham, Yaena Ha, Ickpyo Hong.

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
