## [Decision Letter · Decision Letter 0]

28 Mar 2022

PONE-D-22-03868Item-level Psychometrics of the Ascertain Dementia Eight-Item Informant QuestionnairePLOS ONE

Dear Dr. Hong,

Thank you for submitting your manuscript to PLOS ONE. After careful consideration, we feel that it has merit but does not fully meet PLOS ONE’s publication criteria as it currently stands. Therefore, we invite you to submit a revised version of the manuscript that addresses the points raised during the review process.

ACADEMIC EDITOR: Would be helpful to briefly discuss how the sample size was determined and the sufficiency of the the sample size for CFA. Moreover, in the abstract, you should include actual results from the CFA not just a narrative summary. For instance, you may include the RSMEA and CFI values.

We look forward to receiving your revised manuscript.

Kind regards,

Mohamed F. Jalloh, PhD, MPH

Academic Editor

PLOS ONE

Journal Requirements:

3.Please review your reference list to ensure that it is complete and correct. If you have cited papers that have been retracted, please include the rationale for doing so in the manuscript text, or remove these references and replace them with relevant current references. Any changes to the reference list should be mentioned in the rebuttal letter that accompanies your revised manuscript. If you need to cite a retracted article, indicate the article’s retracted status in the References list and also include a citation and full reference for the retraction notice.

Reviewers' comments:

Reviewer's Responses to Questions

**Comments to the Author**

1. Is the manuscript technically sound, and do the data support the conclusions?

Reviewer #1: Yes

Reviewer #2: Partly

2. Has the statistical analysis been performed appropriately and rigorously? 

Reviewer #1: I Don't Know

Reviewer #2: Yes

3. Have the authors made all data underlying the findings in their manuscript fully available?

Reviewer #1: No

Reviewer #2: Yes

4. Is the manuscript presented in an intelligible fashion and written in standard English?

Reviewer #1: Yes

Reviewer #2: Yes

5. Review Comments to the Author

Reviewer #1: Dear journal editor, thank you for the opportunity. I have to review this manuscript based on my general knowledge. Perhaps it will be nice to show by an analytic expert on tool validation methods.

I hope It will add value for other researchers to address such clinical issues.

Reviewer #2: This is an informative study in an important area of research. Dementia is an illness of great public health concern and as th

e authors have mentioned, early screening results in early management of the illness, and a reduction in family suffering and social costs.

The authors can clarify a few review comments.

1. The data for the analysis was obtained from the NHATs 2019 survey, the authors report on age and sex findings in the results, it is however difficult to understand how heterogenous the study population is, based only on reported age and sex variables.

2. The authors in the results section indicate that out of 4,977 observations only 185 were selected due to missing data. Did the authors analyze if the missingness was at random, is the missing data ignorable? Such high numbers of missing data can be problematic because of bias. This means the results may not be generalizable outside of this study because the data comes from an unrepresentative sample.

3. In the discussion the authors state "According to PubMed, there is an increase in the number of studies related to AD-8,but the item-level psychometrics of this questionnaire remains indeterminate". Can the authors provide a reference for this.

4. In the study limitations the authors state "First, we used secondary data, which may have caused some bias. "Can the authors elaborate on what biases.

5. In the conclusion, the authors need to emphasize that the study substantiate the reliability and validity of the AD-8...in the selected population. Based on the small and likely homogenous sample it may be inaccurate to generalize.

6. One limitation not mentioned by the authors was that criterion validity utilizing diagnostic gold standard and test-retest reliability using repeated measures was not assessed. Perhaps a consideration for future studies.

6. PLOS authors have the option to publish the peer review history of their article (what does this mean?). If published, this will include your full peer review and any attached files.

Reviewer #1: **Yes: **Mohammed Hassen (Assistant professor, Ph.D Fellow in chronic care)

Reviewer #2: No

---

## [Author Response · Author response to Decision Letter 0]

13 May 2022

1. Academic editor:

1) Would be helpful to briefly discuss how the sample size was determined and the sufficiency of the the sample size for CFA. Moreover, in the abstract, you should include actual results from the CFA not just a narrative summary. For instance, you may include the RSMEA and CFI values.

Response: Thank you for considering our research study for publication in Plos One and for helping us make distinct, refined, and effective revisions to our manuscript. In accordance with your recommendations, we have now edited the procedure for selecting an appropriate sample size (Page8-9, Line 252-268). 

In our original manuscript, which made use of the NHATS round 9 data, we included 185 participants as the appropriate sample size to obtain stable item calibration with 99% confidence. However, after reviewing the comments, we have now included 297 additional participants from the recently released NHATS round 10 data. Therefore, a total of 403 participants, including 79 instances of duplicated data, were analyzed in this study, thus allowing for greater item calibration stability. Furthermore, more recent data have now been included in our analysis, and we have thus changed the overall statistical figures. However, our results have not significantly changed. Finally, we have included other CFA-related values, such as RSMEA, CFI, and TLI, in the abstract (Page 2, Line 48-52).

2. Reviewer 1

1) Dear Author, I am not such an expert on such type of tool analysis procedure. But I will try my best to maximize the outcome.

It was better to tell as to why you selected this tool instead of others in your literature part and discussion section. Additionally, it was more strong evidence if you incorporate additional data (Not only 2019 Data Source).

Response: Thank you for the excellent comment. In particular, more participants have been added in the analysis. The original manuscript used data from the NHATS round 9, and 185 participants were selected as the appropriate sample size, offering 99% confidence with stable item calibration with ±0.5 logits. However, after reviewing your comments, we have added 297 more participants from NHATS round 10 data, which was recently released. After merging the two datasets, a total of 403 participants were included in the final analysis, allowing for more robust and precise estimates. Furthermore, we have also included more recent data in our analysis and have thus changed the overall statistical figures. However, our results have not significantly changed compared to before (Page8-9, Line 252-268).

1) Dear author, I just put my minor comments in the main document you can get it. But as for me, I didn’t get any clue why you choose this one?

Response: The AD-8 is a dementia-screening tool demonstrating good reliability and validity. The administration time is less than 3 minutes. The AD-8 test does not require any preparation, and it can be used for screening dementia in a fast and efficient manner without any space and time constraints. Therefore, the AD-8 has merits in the clinical setting; however, the item-level psychometrics have not been conducted. Also, the dichotomous response in the AD-8 is appropriate for Rasch analysis. Therefore, we applied Rasch analysis to confirm the psychometrics of the AD-8.

2) Because I didn’t get access to the full document I failed to see all over model fitness. So could you attach the summary table on your main manuscript and attach to me the raw data to see your model fitness test? 

Response: Thank you for your comment. Please find attached the raw data and correlation table that you requested. An estimator, ML, was not used in our study because of the nature of our data (categorical data). Instead, we have introduced and subsequently applied two appropriate estimators: weighted least squares (WLS) and robust weighted least squares (WLSMV). We described the results of WLSMV in our manuscript based on a previous study (PROMIS study) (Page 10, Line 278-279). 

1) The result of CFA using WLSMV are as follows: = 41.015, df = 20, p = 0.004; CFI = 0.995; TLI = 0.993; RMSEA = 0.051

2) The result of CFA using WLS are as follows: =31.459, df=20, p=0.048; CFI=0.995; TLI=0.993; RMSEA=0.051

Also, our data were extracted from the NHATS, a health-related survey for older adults aged 65 years and above who are Medicare beneficiaries in the United States. Anyone willing to conduct research can download the raw data files after signing up (https://nhats.org/researcher/data-access).

3) Dear Author, if possible, could you please add your structural equation model to the manuscript?

Response: We have now added the diagram of the confirmatory factor analysis model for the AD-8 (Figure 2) (Page 10, Line 285). 

4) If I may not mistake, Rasch analysis was used for categorical data, But the AD-8 rating scale is ordinal data? Could you please clarify this one?

Response: This is an excellent point. As you have mentioned, we used a dichotomous response scale for Rasch analysis. We divided answers into “Yes” or “No” by merging answers such as “No, no change” and “N/A.” We have thus revised the sentence appropriately (Page 5-6, Line 160-166).

5) Dear the author, If I am not get clear on your floor and ceiling effect results, perhaps may need to elaborate more on the result section.

Response: We have explained the criterion of the ceiling and floor effect in the Methods section (Page 8, Line 238-240). Also, we have considered issues related to the floor and ceiling effect in the Discussion section (4th paragraph) (Page 16, Line 55-67).

 

3. Reviewer 2

1) The data for the analysis was obtained from the NHATs 2019 survey, the authors report on age and sex findings in the results, it is however difficult to understand how heterogenous the study population is, based only on reported age and sex variables. 

Response: Thank you for pointing this out. In this study, we hypothesized that the sex and age of the respondents do not affect their responses to every item of the AD-8. To confirm this hypothesis, we performed a differential item functioning analysis. Also, we have now added a demographic table that includes additional participant characteristics shown in Table 1 (Page 9, Line 272). 

2) The authors in the results section indicate that out of 4,977 observations only 185 were selected due to missing data. Did the authors analyze if the missingness was at random, is the missing data ignorable? Such high numbers of missing data can be problematic because of bias. This means the results may not be generalizable outside of this study because the data comes from an unrepresentative sample.

Response: Thank you for this comment. We do agree that the sentence we described in the “participants” session could be confusing for the reader. Not all respondents in the NHATS participated in the AD8 questionnaire. More specifically, only individuals with cognitive problems reported by their proxy participated in the AD-8. Among the 4,977 observations in the NHATS round 9 databases, 273 participants responded to the AD-8 questionnaire. After excluding 88 observations with missing data, a total of 185 responses to these eight questions were selected. In addition, we have included the recent NHATS round 10 dataset, thus adding more participants and strengthening the respective evidence. We have also refined and further clarified the data collection process (Page8-9, Line 252-268). In addition, we have now mentioned the limitation caused by the missing observations in the discussion section (Page 17, Line 95-97).

3) In the discussion the authors state "According to PubMed, there is an increase in the number of studies related to AD-8,but the item-level psychometrics of this questionnaire remains indeterminate". Can the authors provide a reference for this. 

Response: We appreciate your comment. We have investigated research trends on AD8 using the PubMed search engine. Therefore, this sentence has not been sourced from other studies but it is the result of our research process. To clarify this, we have revised the sentence (Page 15, Line 28-30).

4) In the study limitations the authors state "First, we used secondary data, which may have caused some bias. "Can the authors elaborate on what biases.

Response: Thank you for the comment. Accordingly, we have now listed potential sources of biases that might have been caused by the use of secondary data, under limitations (Page 17, Line 93). 

5) In the conclusion, the authors need to emphasize that the study substantiate the reliability and validity of the AD-8...in the selected population. Based on the small and likely homogenous sample it may be inaccurate to generalize. 

Response: We agree with your point. The use of the term “substantiate” in these cohort datasets is inappropriate. To clarify the same, we have specified the population and revised the sentence accordingly. (Page 18, Line 112-114).

6) One limitation not mentioned by the authors was that criterion validity utilizing diagnostic gold standard and test-retest reliability using repeated measures was not assessed. Perhaps a consideration for future studies. 

Response: We appreciate your comment. We have now added this information as a limitation of our study, as recommended by you (Page 17-18, Line 107-109).

---

## [Decision Letter · Decision Letter 1]

7 Jun 2022

Item-level Psychometrics of the Ascertain Dementia Eight-Item Informant Questionnaire

PONE-D-22-03868R1

Dear Dr. Hong,

We’re pleased to inform you that your manuscript has been judged scientifically suitable for publication and will be formally accepted for publication once it meets all outstanding technical requirements.

Kind regards,

Mohamed F. Jalloh, PhD, MPH

Academic Editor

PLOS ONE

Additional Editor Comments (optional):

Reviewers' comments:

Reviewer's Responses to Questions

**Comments to the Author**

1. If the authors have adequately addressed your comments raised in a previous round of review and you feel that this manuscript is now acceptable for publication, you may indicate that here to bypass the “Comments to the Author” section, enter your conflict of interest statement in the “Confidential to Editor” section, and submit your "Accept" recommendation.

Reviewer #1: All comments have been addressed

2. Is the manuscript technically sound, and do the data support the conclusions?

Reviewer #1: Yes

3. Has the statistical analysis been performed appropriately and rigorously? 

Reviewer #1: Yes

4. Have the authors made all data underlying the findings in their manuscript fully available?

Reviewer #1: Yes

5. Is the manuscript presented in an intelligible fashion and written in standard English?

Reviewer #1: Yes

6. Review Comments to the Author

Reviewer #1: The paper was much matured than the previous submission. I think it is sound for publication.

Thank you

7. PLOS authors have the option to publish the peer review history of their article (what does this mean?). If published, this will include your full peer review and any attached files.

Reviewer #1: **Yes: **Mohammed Hassen Salih

---

## [Editor Report · Acceptance letter]

10 Jun 2022

PONE-D-22-03868R1 

Item-level Psychometrics of the Ascertain Dementia Eight-Item Informant Questionnaire 

Dear Dr. Hong:

I'm pleased to inform you that your manuscript has been deemed suitable for publication in PLOS ONE. Congratulations! Your manuscript is now with our production department. 

Kind regards, 

on behalf of

Dr. Mohamed F. Jalloh 

Academic Editor

PLOS ONE